# Critical Salt Loading in Flexible Poly(vinyl alcohol) Sensors Fabricated by an Inkjet Printing and Plasma Reduction Method

**DOI:** 10.3390/mi13091437

**Published:** 2022-08-31

**Authors:** Evan Chou, Yongkun Sui, Hao Chong, Christina Brancel, John J. Lewandowski, Christian A. Zorman, Gary E. Wnek

**Affiliations:** 1Department of Macromolecular Science and Engineering, Case Western Reserve University, Cleveland, OH 44106, USA; 2Department of Electrical, Systems and Computer Engineering, Case Western Reserve University, Cleveland, OH 44106, USA; 3Department of Chemistry, Case Western Reserve University, Cleveland, OH 44106, USA; 4Department of Materials Science and Engineering, Case Western Reserve University, Cleveland, OH 44106, USA

**Keywords:** flexible printed sensors, plasma sintering, salt interface

## Abstract

We report a low-temperature inkjet printing and plasma treatment method using silver nitrate ink that allows the fabrication of conductive silver traces on poly(vinyl alcohol) (PVA) film with good fidelity and without degrading the polymer substrate. In doing so, we also identify a critical salt loading in the film that is necessary to prevent the polymer from reacting with the silver nitrate-based ink, which improves the resolution of the silver trace while simultaneously lowering its sheet resistance. Silver lines printed on PVA film using this method have sheet resistances of around 0.2 Ω/□ under wet/dry and stretched/unstretched conditions, while PVA films without prior treatment double in sheet resistance upon wetting or stretching the substrate. This low resistance of printed lines on salt-treated films can be preserved under multiple bending cycles of 0–90° and stretching cycles of 0–6% strain if the polymer is prestretched prior to inkjet printing.

## 1. Introduction

The field of flexible electronics has always revolved around the challenge of combining electrical and flexible components without compromising the performance of either [1]. Highly conductive materials such as metals often lose their conductivity upon stretching or are too stiff to stretch altogether [2]. One strategy to address this issue is to incorporate conductive nanofillers into a polymer matrix [1,3]. A variety of conductive fillers and polymer matrices can be used [2,4,5], with the most common being carbon black dispersed in poly(dimethylsiloxane) (PDMS) [3]. Such composite materials have both adequate conductivity and flexibility but excel at neither; the conductive material being loaded inherently decreases the flexibility of the polymer substrate, while the substrate acts as a source of electrical insulation to reduce the conductivity of the filler [6]. Alternate strategies for fabrication of flexible electronics involve sacrificial transfer of conductive patterns, such as metal nanowires onto a flexible substrate [7,8], as well as printing nanoparticle inks onto the substrate and then sintering [9]. Metal nanowires are costly to make and require multiple transfer steps to achieve suitable conductivity and mechanical robustness [10]. Meanwhile, metal nanoparticle inks are difficult to process due to agglomeration [5], while also needing to be sintered in order to demonstrate their excellent electrical conductivity. Sintering requires very high temperatures at which most polymeric substrates will degrade [9]. For flexible electronics to gain more widespread use, there would need to be a process that allows for a variety of substrates, generates films with excellent electrical conductivity and flexibility, utilizes low-cost materials, and is simple to fabricate.

One such process was recently developed by members of our team using an inkjet printing and plasma treatment method to convert silver nitrate ink into metallic silver traces [11]. Unlike the nanoparticle ink method, it produces conductive silver lines at relatively low temperatures, while also avoiding the issue of nanoparticle aggregation since it avoids the use of metal nanoparticles altogether. The resistivity of the printed silver lines is extremely low, comparable to that of bulk silver, making it an attractive option for high-end sensing applications. Moreover, the relative simplicity of this approach may allow for scalable fabrication of complex patterns with high resolution. As a result, many potential substrate materials may become viable.

One in particular, poly(vinyl alcohol) or PVA, is attractive due to its hydrophilicity, skin-like modulus when wet, biocompatibility, and ability to be easily solvent cast from water [12]. It is worthwhile to summarize a few salient points about PVA. While the acronym ‘PVA’ suggests that the polymer is made by polymerization of vinyl alcohol, that is not the case as vinyl alcohol does not exist in any appreciable amount since it is in equilibrium with the highly thermodynamically favored tautomer acetaldehyde. Therefore, PVA must be prepared by an alternate route, which commercially involves the synthesis of poly(vinyl acetate) followed by hydrolysis (or alcoholysis) to PVA. Importantly, this route leads to many grades of PVA based on the degree of hydrolysis, which can range from about 86% to 99+%. It is the most highly hydrolyzed version that we use (99+%, sometimes referred to as ‘super-hydrolyzed’ [12] and which will be simply referred to as PVA in this paper) as the backbone purity promotes partial crystallization, even in the absence of any appreciable tacticity of the PVA backbones, upon casting from aqueous solution followed by ambient drying. This affords a water-swellable but water-insoluble film. Thermal degradation of PVA starts around 250 °C [13], while traditional sintering goes up to 300 °C [14], meaning the low-temperature approach is particularly useful for PVA as a substrate for electronics.

Despite the apparent advantages, preliminary results showed that silver lines printed on PVA through the inkjet printing method actually had higher resistances and lower print resolutions compared to the same silver lines printed on other common substrates, such as polyimide, polyethylene, and polyethylene terephthalate [11]. There was also a visible “staining” effect on these printed lines for PVA, in which discoloration would occur on the polymer shortly after printing and smear outward from the silver. This observation led us to believe staining is due to the reaction of PVA with silver, producing silver nanoparticles that are free to diffuse throughout the polymer. For silver nitrate to be converted to metallic silver, there needs to be little to no interaction of the PVA with silver. For this reason, we incorporated a small quantity of magnesium chloride salt into the casting solution and found that the resulting film was able to produce lines with improved conductivity and resolution. We propose that magnesium salt preferentially reacts with silver nitrate ink before it reaches the PVA, forming silver chloride crystals. These crystals prevent any remaining silver ink from reacting with the PVA, leaving them to be completely converted to metallic silver. Furthermore, we adopt a strategy for retaining low electrical resistance upon subsequent stretching by first prestretching the polymer film before inkjet printing. With this method, films of printed silver on PVA retain their low resistance values even under bending and stretching cycles, showing that modified PVA films can be used to further broaden the scope of the flexible electronics field.

## 2. Materials and Methods

The PVA powder used was Elvanol™ 90-50 (99+% hydrolyzed, 35 kDa), a gift from Kuraray. Magnesium chloride hexahydrate, zinc chloride, calcium chloride, and silver nitrate salts were all purchased from Sigma-Aldrich, Burlington, MA, USA.

### 2.1. Solvent Casting of Poly(Vinyl Alcohol) Films and Their Derivatives

A 3–5 wt% PVA casting solution was prepared by slowly dissolving 3 g increments of PVA powder in 95 °C deionized water under vigorous stirring. Upon complete dissolution, the solution was cooled to 60 °C to prepare for storage. If the PVA remained fully dissolved, it was transferred at this temperature into a capped glass vessel. Otherwise, the solution was heated back up to 95 °C to dissolve any remaining solids, then cooled to 60 °C, and this step was repeated until a homogeneous solution was achieved. The prepared PVA stock solution was used to cast films in plastic Petri dishes (Figure 1).

At least 15 mL of this solution per film or a corresponding thickness of 60 µm was necessary for suitable mechanical properties and a uniform surface when printing. Between 1 and 15 mL of deionized water was added to the dish containing PVA solution and gently stirred. The resulting solution was cast at room temperature for 1–2 days or until fully dried. These conditions produced films with a more uniform surface compared to drying in an oven at elevated temperatures. Before printing, the films were dried while held taut for at least one hour in a 60 °C oven to prevent inconsistencies due to humidity as well as wrinkling.

A stock solution of 0.05 M salt in deionized water was prepared for the salt-loaded samples. The divalent salts tested were magnesium chloride hexahydrate, calcium chloride, and zinc chloride. The monovalent salts lithium chloride, potassium chloride, and sodium chloride were also tested but did not display suitable mechanical or optical properties for use in flexible electronics. Between 1 and 15 mL of the prepared salt solution was added to a Petri dish containing PVA and gently stirred. The resulting solution was dried according to the same method as the neat PVA samples.

### 2.2. Mechanical Testing of Cast PVA Films

The PVA films were cut into dog bone-shaped specimens in accordance with ASTM D638-14, which is the standard test method for tensile properties of plastics (Figure 2). G stands for gauge length, L is the length of the narrow section, D is the distance between grips, LO is the overall length, W is the width of the narrow section, Wc is the width at the center, WO is the overall width, R is the radius of fillet, and T is the thickness.

The dimensions of the specimens are listed in Table 1. All specimens were cut by hand using a sharp x-acto™ blade. Tensile testing of the prepared specimens was conducted using the tabletop Instron 1026 machine in the Advanced Manufacturing and Mechanical Reliability Center (AMMRC: https://ammrc.case.edu/) at CWRU. Tensile testing was conducted at two strain rates (1.1 × 10^−2^/s and 2.2 × 10^−2^/s). These strain rates were achieved by conducting the tests at a displacement rate of either 5 mm/min or 10 mm/min, respectively. The tensile tests were conducted under three conditions for each strain rate in order to determine the impact of humidity on the mechanical properties: low-humidity environment (<15% RH), high-humidity environment (50% RH), and presoaking the specimens in water for 24+ h prior to testing in a high-humidity environment (i.e., 50% RH). High-humidity environments were achieved by using a humidifier to increase the humidity of the area immediately surrounding the specimens during testing. The humidifier was aimed to puff water vapor directly onto the specimens during testing. Using a hygrometer, the humidity of the testing environment near the specimen surface was recorded for each specimen.

### 2.3. Experimental Test Methods

#### 2.3.1. Drop Casting

The prepared silver nitrate ink (30 μL, 0.03 mmol) was added to the dried prepared film using a micropipette. The ink was left to sit on the substrate for thirty minutes. The staining of the film and spreading of the ink were recorded in increments of 10 min, 20 min, and 30 min.

#### 2.3.2. Inkjet Printing

We prepared 1.5 M silver nitrate ink according to the following procedure: 20 mL of ethylene glycol, 8 mL of ethanol, 7 mL of deionized water, and 12.7 g of silver nitrate salt were mixed at 21.5 °C. This ratio of ethylene glycol and ethanol determines the viscosity and surface tension of the printed droplets, which in turn is related to the final print resolution [11].

The inkjet printer used was a Dimatix DMP3000 (Fujifilm Dimatix, Lebanon NH, USA) with 16 piezoelectric nozzles.

Simple designs (see Figure 3) measuring roughly 1 cm by 0.2 cm were printed with a 50 °C printer head on a room-temperature printer bed. Drop spacing was 15 μm.

#### 2.3.3. Plasma Treatment

The plasma chamber used was a March PX250 (Nordson Electronics Solutions, Westlake OH, USA), which was connected to an external 300 W, 13.56 MHz RF power supply (ENI, ACG-6B-06).

Prior to inkjet printing, samples were treated with oxygen gas in an evacuated plasma treatment chamber to remove any surface contaminants. This treatment was performed at 60 W for 1 min. After inkjet printing, printed samples were treated with pure argon gas (99.9999%, Air Gas) in the same evacuated plasma treatment chamber at 300 W for 10 min at a pressure of roughly 630 mTorr. After plasma treatment, sample resistance was measured using a standard multimeter to make sure the print was successful. Sheet resistance was measured with a four-point probe [16] in order to compare our samples with others in the literature.

## 3. Results

### 3.1. Silver Nitrate Staining of PVA Film

Inkjet printing of electronic materials is a process by which droplets of ink are converted to conductive traces on a substrate surface [17]. During this process, the substrate–ink interaction is critical. For this study, silver was chosen as the metal ink due to its high electrical conductivity and excellent physical properties [5]. However, it also suffers from high reactivity with most polymeric materials. This leads to a type of “staining” phenomenon as the ink comes in contact with polymers such as PVA, causing the surface of the polymer to become discolored due to reacted silver particles [18]. These particles can lead to an overall loss of conductivity or even prevent further printing entirely.

However, by incorporating a small quantity of inorganic salt such as magnesium chloride into the PVA polymer matrix, we found that we can provide an alternate reaction pathway and prevent staining entirely. In this system, silver nitrate ink preferentially reacts with embedded magnesium chloride crystals, forming a barrier layer of silver chloride on the surface. Further additions of silver ink during the printing process will rest on top of the silver chloride layer and be converted to metallic silver upon plasma treatment. Modification of the substrate by low salt loading provides a cheap and effective method for tuning the surface reactivity of a hydrophilic substrate like PVA. A diagram of the proposed mechanism is shown in Figure 4. Silver nitrate ink preferentially reacts with the magnesium chloride salt, forming a silver chloride precipitate. The silver chloride layer acts as a barrier preventing the rest of the ink from reacting with PVA.

To establish a fundamental framework for this reaction, we first mixed a drop of silver nitrate ink into solutions of PVA and PVA mixed with magnesium chloride salt and let these sit for 2 days. The sample with PVA and magnesium salt immediately formed a white precipitate of silver chloride upon addition of silver ink, which then settled to the bottom of the vial. Meanwhile, the sample with no added salt turned homogeneously brown over the course of two days to indicate the formation of silver nanoparticles. These observations in the liquid state point to the preferential reaction of silver–magnesium over silver–PVA, which we later tested in the solid state to see if the same principles apply. By altering the concentration of salt over a wide range, we saw a transition between forming a white precipitate and forming a brown silver nanoparticle solution, leading to the idea that perhaps a critical threshold is involved, as shown in Figure 5.

Specifically, the 3:1 volume mixture of PVA:Mg appeared to be the transition mixture, corresponding to approximately a stoichiometric conversion of silver by magnesium. Any addition of magnesium below that threshold yields silver nanoparticle staining, whereas any amount greater than the threshold is free of nanoparticle contamination.

To verify this mechanism in the solid state, the composition of the barrier layer was analyzed via 2D X-ray diffraction (Bruker Discover D8 with VANTEC-500 solid state detector, Billerica, MA, USA) to check for silver chloride on the PVA surface, as shown in Figure 6 below.

We used 2D XRD to determine if AgCl crystals are present on the surface of magnesium-loaded PVA upon depositing the ink and before plasma treatment. To emulate the same conditions, we deposited a drop of silver nitrate ink onto the surface of the PVA until it turned cloudy, then removed the excess water and allowed the surface to dry. AgCl XRD peaks are known to be relatively sharp and distinct, but this approach failed to account for the difficulty of obtaining a clean XRD on a nonideal substrate like the one we use to describe our system. Standard XRD procedure assumes a very flat and uniform sample, while the distribution of MgCl_2_ and presumably AgCl throughout the polymer is random. Therefore, the XRD also includes the PVA surface which is quite noisy due to the semicrystalline nature of PVA [20]. Furthermore, it was not possible to subtract a background spectrum of PVA because this would mean comparing two different films. Because of these complications, we focused on the likelihood of each form of silver being present on the polymer surface based on possible reactions and the spectrum peaks. As shown in green, there are defined peaks in all of the areas that indicate AgCl, while other forms of silver have peaks that are not shown on the spectrum or are missing peaks shown on the spectrum. Based on all of these factors, we can reasonably conclude that the crystals bound to the surface of the PVA film are AgCl crystals.

The formation of silver chloride crystals in the magnesium-loaded PVA leads to a drastic reduction in electrical resistivity and improved print resolution compared to silver lines printed on neat PVA films. Table 2 shows the resistance values under wet conditions of silver lines printed on PVA with varying degrees of salt loading. Resistance was measured using a standard multimeter. The probes were connected to a thin metal wire, which was then connected to the silver trace using a small amount of silver paste. Again, there appears to be a transition region where added silver nitrate is stoichiometrically converted by magnesium chloride, which corresponds to a drastic dip in electrical resistance of the printed lines.

Compositions are written as [volume of stock PVA in mL]-[volume of stock magnesium salt solution in mL]. Stock PVA was 5% by weight, while stock magnesium solution was 0.05 M. The bolded composition indicates where the resistance is drastically reduced.

Using the same procedure with different divalent salts such as calcium chloride and zinc chloride did not seem to affect the print quality or electrical resistance, indicating that it is the formation of silver chloride alone that leads to this phenomenon. However, the choice of salt does dictate the mechanical, thermal, and optical properties of the PVA film [13,21,22].

### 3.2. Inkjet Printing and Plasma Treatment

Pretreatment of the prepared films with oxygen gas plasma was necessary to ensure the film surface was free from debris and contaminants. Pretreatment also primed the surface to bond with the silver nitrate ink. If samples were not pretreated, print resolution was compromised and there was no visible formation of silver chloride upon printing. Upon Ar plasma treatment, the silver was still reduced, but the printed lines had resistance values greater than 1 kΩ, which is much too high for most electronic and sensing applications.

Print quality, measured qualitatively by observing resolution, was dependent on a number of parameters: film water content, print bed temperature, ink concentration, and salt concentration. Though all samples were dried after preparation, the films may have reabsorbed some moisture in between sample preparation and printing. In the presence of excess water, an accelerated brown staining reaction was observed. To keep all conditions consistent, we dried all samples in an oven at 60 °C for one hour immediately prior to printing. Likewise, a heated printer platen also accelerated the brown staining reaction. For normal printing operations, the platen is kept around 50 °C, but this temperature is unsuitable for our process because it softens the polymer substrate and encourages staining. Instead, the printer platen was kept at room temperature when printing; care must be taken to ensure the platen remains cool, as it gradually warms up when printing multiple samples in a row.

Ink concentration was important in ensuring there was enough silver present to produce conductive lines. If the ink concentration was too low, e.g., 1 M, the silver lines were dull and had high resistance values after Ar plasma treatment (see Figure 7a). When the ink concentration was increased to 1.5 M, the lines had better resolution and the characteristic shine of metallic silver.

Salt concentration was another determining factor in print quality; in the presence of a salt, the PVA films visually produced cleaner, higher-resolution prints with less staining than what we saw in neat PVA (see Figure 7b). These results are in accordance with the proposed silver chloride mechanism.

### 3.3. Mechanical and Electrical Properties

As mentioned previously, upon wetting, the resistance of a silver trace printed on salt-loaded PVA was drastically lower than that of a trace printed on neat PVA. The difference was approximately two to three orders of magnitude.

For use in flexible electronics, these printed lines must retain their conductivity under stretching and bending cycles. This is difficult to achieve because silver metal is intrinsically rigid, making it difficult to stretch without forming cracks. In recent years, a common strategy to retain the conductivity of printed metal lines is to print on a “prestretched” substrate [23]. For a hydrophilic substrate like PVA, this can be achieved by first softening with water, as shown below, and prestretching, allowing the film to dry in this stretched state, then printing on it.

Figure 8 shows the significant and beneficial effects of preconditioning (i.e., soaking) and the test environment (i.e., RH) on the mechanical response of PVA. Figure 8a shows the consistently high stiffness/strength and lower strain to failure for PVA samples tested under dry (i.e., <15% RH) conditions. In contrast, Figure 8b shows the extreme effects of preconditioning (i.e., soaking) and subsequent testing at RH = 50% on stiffness/strength, namely significant and beneficial reductions, which accompany the even greater increases in strain to failure. Similar results were obtained for samples tested at the higher strain rate as shown in Figure 9 and Figure 10 which summarize these dramatic effects for the conditions tested. The samples have slight variations in thickness dimension and structure because the dog bones were cut from different areas of the same polymer sheet, and solvent casting of PVA leads to slight variations in film thickness.

Because the metal ink was printed and plasma-converted while the film was under tension, when the tension is released, the silver trace will compress. Compression does not significantly impact resistance compared to tension [24], so the silver lines can retain their low resistance in both the dry (stretched) and wet (unstretched) state.

For comparison with literature values, the same process was repeated, except this time for sheet resistance. Sheet resistance (SR) was measured on PVA films with printed silver squares according to the standard four-point probe method [16]. To ensure that the spatial distribution of silver did not affect our measurement, we oriented the probe along three directions on the silver square: length (L), width (W), and diagonal (D). The results are reported in Table 3 below. For comparison, reference values from related literature [25,26] are shown as well, in Table 4.

After ensuring that each sample was dried the same way before printing and plasma treatment under the same conditions, we found that the sheet resistance of printed silver on two Mg-loaded PVA samples had negligible variation (0.2746 Ω/□ vs. 0.2879 Ω/□), therefore the resistivity is also relatively consistent. Because sheet resistance measurements are more time-intensive, we mostly report plain resistance values for comparison in this work and found no significant variation in the printed lines of similar dimensions (~6–12 Ω) in at least 10 tested samples using a standard multimeter and thin metal wires/silver paste. These values align with our group’s previously reported results on less reactive substrates such as PET, Kapton, printing paper, etc., with all other conditions held constant.

Sheet resistance is proportional to resistivity and is used for the sake of comparison by assuming that the thickness of the printed silver patterns remains within the expected range for inkjet-printed electronics. Our team has previously measured print thicknesses for a number of common metals including the silver ink used in this work [11]. However, it is difficult to accurately measure thickness and sheet resistance simultaneously while keeping other factors such as substrate, wetting, and strain constant. Analysis of the printed silver resistivity as a function of wetting and stretching will be the topic of future work. Scanning electron microscopy on environmentally conditioned PVA samples may provide better insight on how the structure of the printed silver changes in response to factors such as strain and moisture.

Because of the prestretching, our reported resistance values do not drastically change upon repeated wetting and stretching cycles. However, the amount of water added in order to stretch the substrate must be carefully controlled to prevent overloading of the polymer matrix. Complete soaking leads to irreversible physical deformation, which impacts both the mechanical properties of the film and the electrical properties of the silver trace. Resistance of the printed silver lines over multiple stretching cycles was measured by securing the wet films on a crank-operated tensile apparatus and stretching them from 0–6% strain, which appeared to be the upper limit for these thin films. The results are shown in Table 5 below for a prestretched PVA-Mg film.

We chose to start with a low N cycle number for the wet strain measurements. Drift will most likely occur at higher N and where it occurs can provide valuable insight on how the silver chloride interface affects the device. However, it has been difficult to automate this process to extend to higher cycle numbers due to the difficulty of maintaining the same wet and dry states of the polymer while simultaneously measuring electrical resistance of a wet sample (maintaining good contact with the probes) with reasonable precision. We hope to address this concern in future work by coupling a cyclic moisture apparatus with a cyclic strain device and making sure the strain/wetting does not interfere with electrical probe contact.

Furthermore, the dependence of resistance on wet contraction of the polymer substrate can be measured in real time by fixing thin metal wires to the silver contact points.

The results of this test are plotted in Figure 11 for prestretched Mg-loaded PVA. We demonstrate that, even though the stress is increasing significantly due to the contraction of the polymer during the drying process, the resistance of the printed silver lines remains largely unchanged. As expected, contraction does not significantly affect the resistance, which means that prestretched samples can maintain their high conductivity within a range defined by the degree of initial strain during printing.

Bending studies were conducted on the same apparatus used to generate Figure 11. However, instead of allowing the films to dry under tension, the ends were forcibly compressed in order to monitor a range of bending angles. The dependence of resistance on bending angle is shown in Figure 12 for the range 0–90°.

## 4. Discussion

We can conclude that our salted PVA films maintain excellent conductivity in the range of 0–90° for bending and at least 0–6% for strain, are robust to physical contact with moisture, and can act as a stable platform for high-resolution traces by a low-cost, low-temperature inkjet printing method. Unlike other stretchable sensors, in which the increase in resistance due to strain is the objective [27], our design enables the sensing of temperature, for example, that is independent of the stress state and moisture state of the polymer. Other studies have reported high selectivity for resistance changes due to temperature over strain [28] or pressure over strain [29], but few address the complications of hydrophilic polymers and moisture. Further investigation will focus on different choices of substrate, ink, and sensing applications, while still emphasizing the importance of low cost and environmental robustness. In particular, graphene oxide [30] or graphene hybrids [31] are an attractive option due to their superior flexibility compared to silver, which leads to an increase in the maximum allowable strain of the sensing devices. With electronics constantly adapting to the needs of the modern world, the possibility of printing electronics onto any hydrophilic substrate or even directly onto skin itself [32] lends itself to a variety of applications, such as medical diagnostics and robotics, that require a sensing component.

## Figures and Tables

**Figure 1 micromachines-13-01437-f001:**
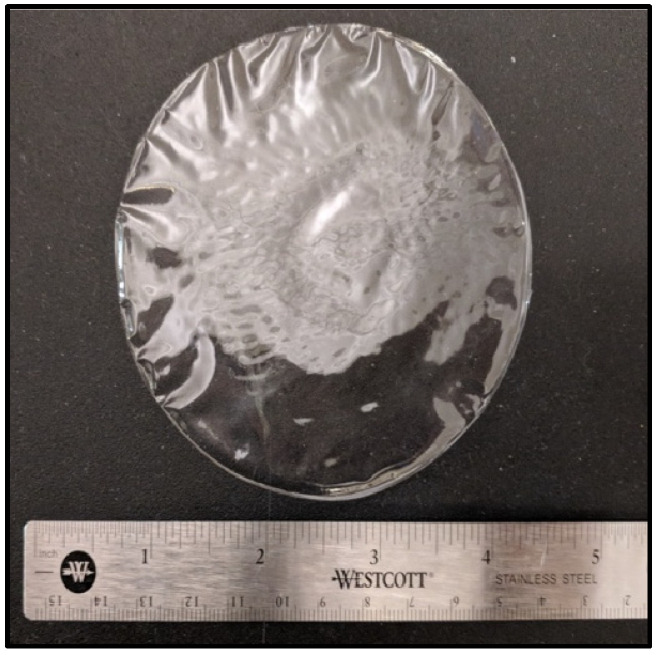
Image of PVA film.

**Figure 2 micromachines-13-01437-f002:**
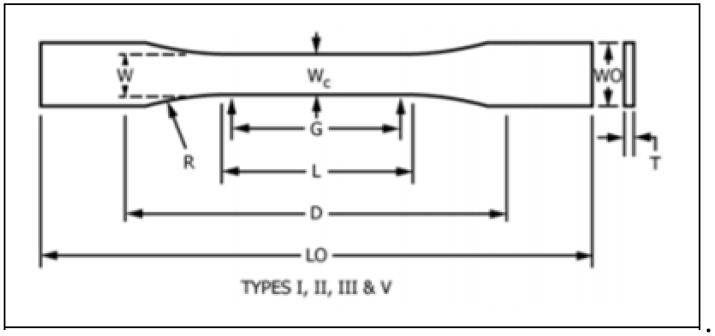
Diagram of ASTM standard tensile specimen [15].

**Figure 3 micromachines-13-01437-f003:**
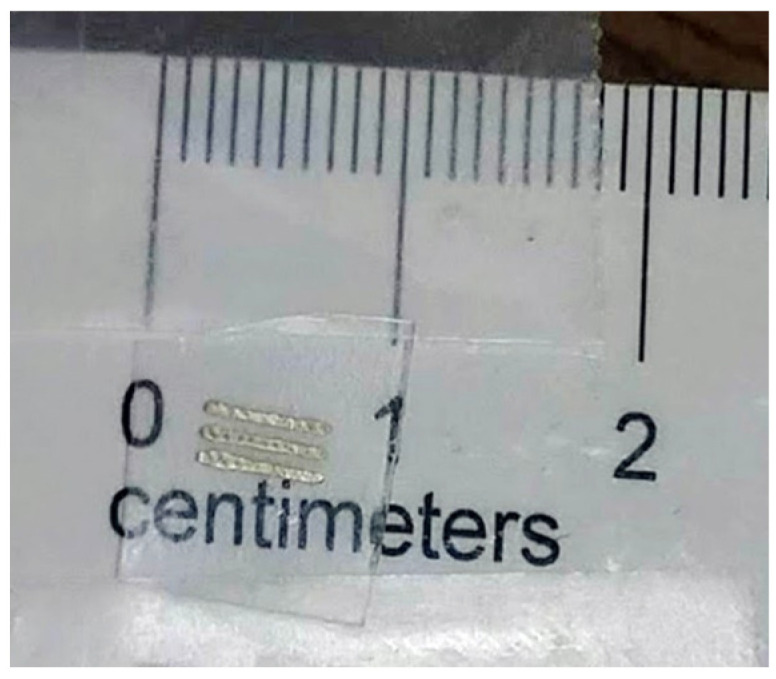
Simple straight-line pattern of silver after inkjet printing and plasma treatment.

**Figure 4 micromachines-13-01437-f004:**
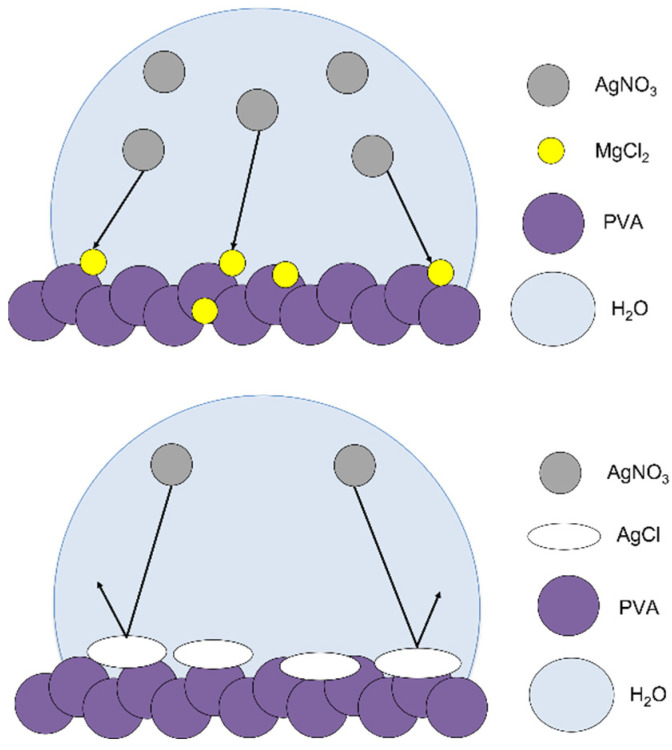
Proposed mechanism of loaded salt as a barrier to silver staining on PVA films.

**Figure 5 micromachines-13-01437-f005:**
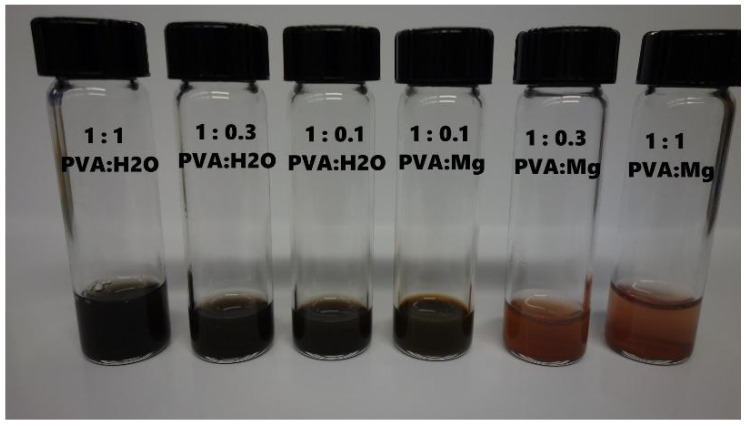
Visual depiction of PVA–water and PVA–salt solutions in order of increasing salt concentration from left to right.

**Figure 6 micromachines-13-01437-f006:**
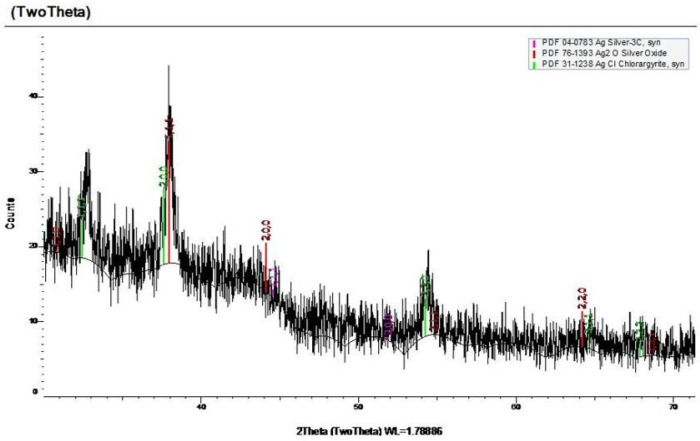
XRD of MgCl2-loaded PVA (3:1 ratio of PVA:MgCl2). Appropriate reference peaks are displayed and show good overlap with the sample spectrum [19].

**Figure 7 micromachines-13-01437-f007:**
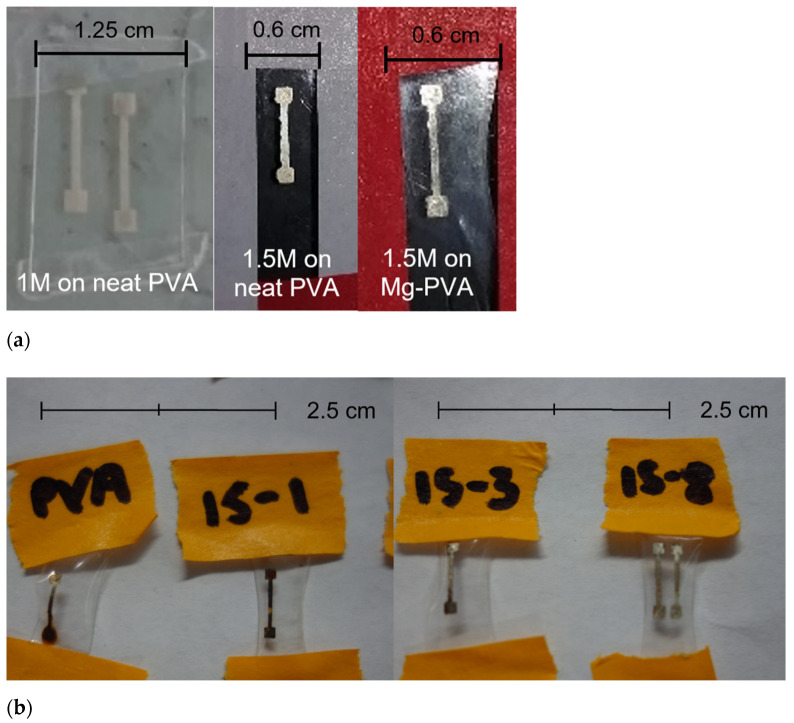
(**a**) From left to right, neat PVA printed with 1 M silver ink, neat PVA printed with 1.5 M ink, and Mg-loaded PVA printed with 1.5 M ink. (**b**) Silver lines printed on PVA loaded with varying concentrations of salt, under the same notation as before.

**Figure 8 micromachines-13-01437-f008:**
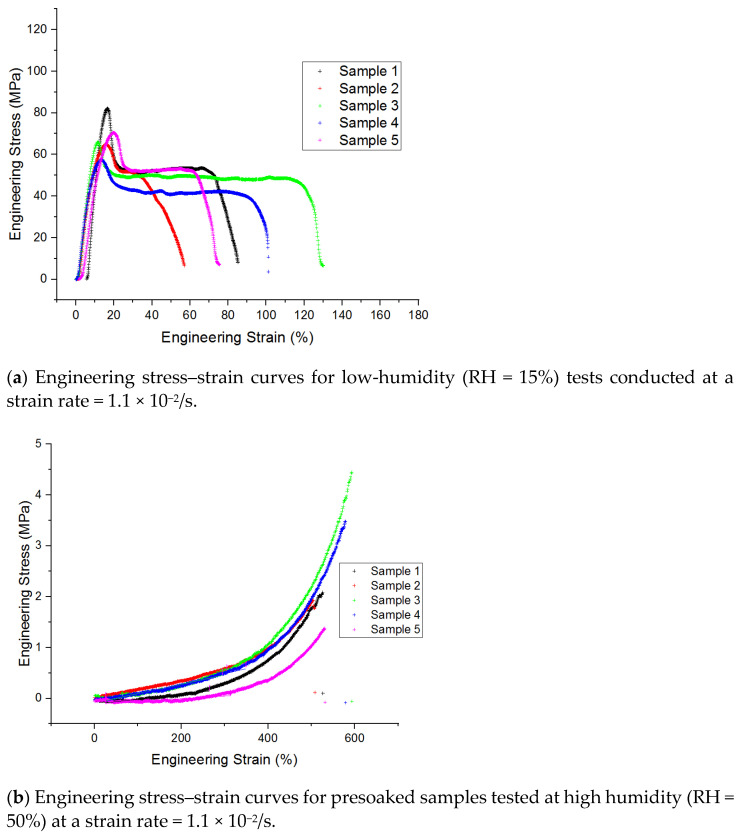
Change in stiffness/modulus and characteristic stress-strain curve upon wetting fully hydrolyzed PVA films. Semi-crystalline behavior is observed in the dry samples (**a**) tested at low humidity (<15% RH) while a J-curve reminiscent of biological tissue or a prestressed elastomer is observed in the presoaked samples (**b**) tested at high humidity (50% RH).

**Figure 9 micromachines-13-01437-f009:**
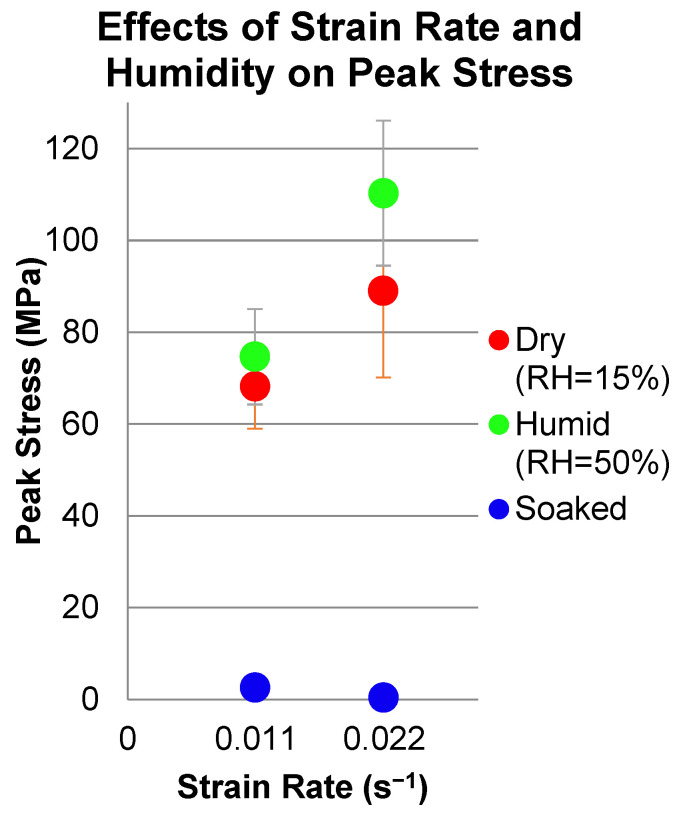
Summary of average peak stress versus strain rate for PVA conditions tested. Soaked samples tested at RH = 50%.

**Figure 10 micromachines-13-01437-f010:**
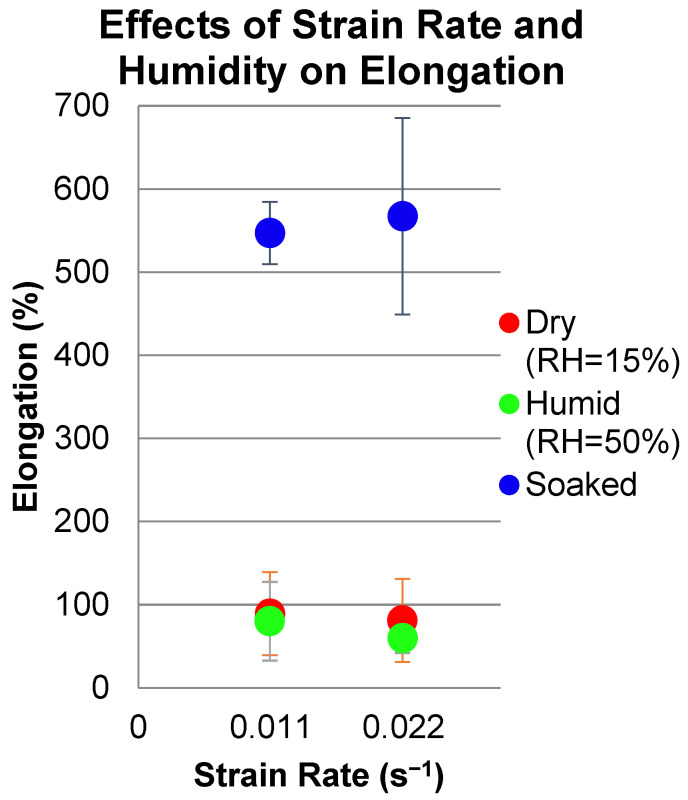
Summary of average elongation versus strain rate for PVA conditions tested. Soaked samples tested at RH = 50%.

**Figure 11 micromachines-13-01437-f011:**
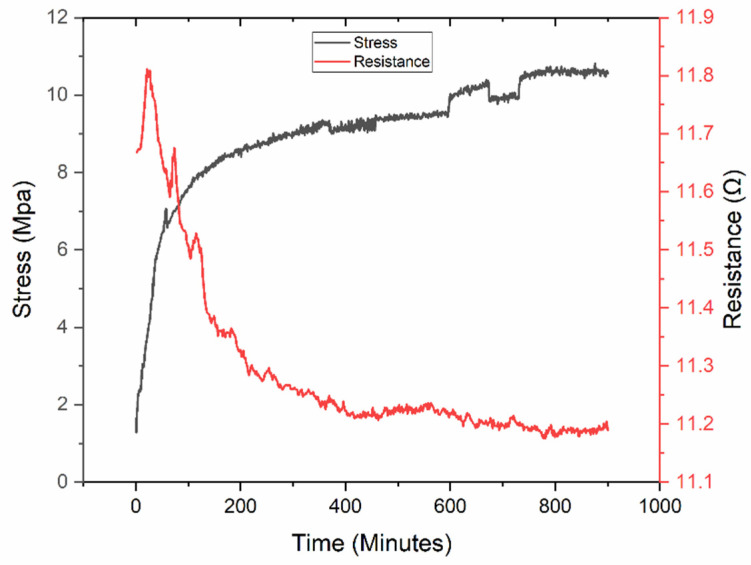
Wet contraction test for strain (black, left) and resistance (red, right). Strain increases significantly due to the removal of water and polymer stiffening, while electrical resistance remains fairly constant.

**Figure 12 micromachines-13-01437-f012:**
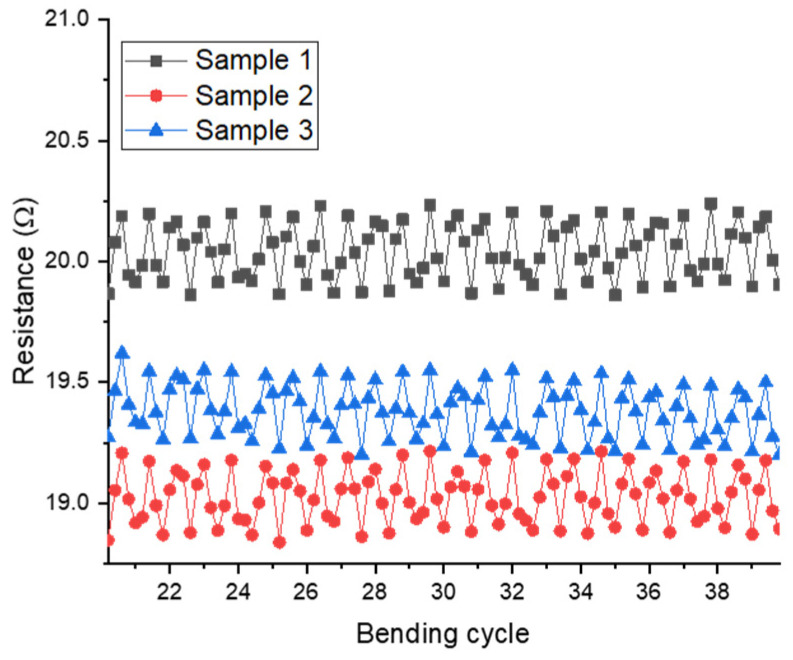
Bending test for resistance of silver trace printed on dry PVA film, in which each cycle goes from 0° → 50° → 90° → 50° → 0°. Bending within this range does not appear to lead to any significant loss of conductivity, even after multiple cycles.

**Table 1 micromachines-13-01437-t001:** PVA Sample Measurements and Test Conditions.

Sample	Thickness (mm)	Width (mm)	Length (mm)
2 (dry: <15% RH)	0.11	3.10	7.62
4 (humid: 50% RH)	0.11	3.53	7.62
6 (soaked + 50% RH)	0.99	4.83	18.0

**Table 2 micromachines-13-01437-t002:** Electrical resistance of printed silver as a function of magnesium salt present in PVA solution prior to casting.

Composition	Resistance (kΩ)
Stock PVA	3.64
15-1	7.00
**15-3**	**0.015**
15-8	0.020

**Table 3 micromachines-13-01437-t003:** Sheet resistance comparison between silver lines printed on unstretched and prestretched PVA and its salt-loaded derivative.

Sample	SR_L_ (Ω/□)	SR_W_ (Ω/□)	SR_D_ (Ω/□)
Prestretched PVA	0.4406	0.5126	0.7512
Unstretched PVA	0.2635		
Prestretched PVA-Mg	0.2746	0.2170	0.2475
Unstretched PVA-Mg	0.2024	0.2139	0.2073

**Table 4 micromachines-13-01437-t004:** Reported sheet resistance values for flexible printed sensors from relevant literature.

Literature References	SR (Ω/□)
Inkjet graphene biosensor [25]	110
Silver on PET [26]	0.95 (light anneal), 2.03 (thermal anneal)

**Table 5 micromachines-13-01437-t005:** Resistance of PVA printed silver over three wet strain cycles.

	Resistance at 0% Strain (Ω)	Resistance at 6% Strain (Ω)
Cycle 1	7.0	8.9
Cycle 2	8.3	8.9
Cycle 3	8.3	8.9

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
