# Peer review of "Critical Salt Loading in Flexible Poly(vinyl alcohol) Sensors Fabricated by an Inkjet Printing and Plasma Reduction Method"

_micromachines, 2022, doi:10.3390/mi13091437_

Round 1

Reviewer 1 Report

This paper introduced a method to fabricate the sensors by the low-temperature inkjet-printing and plasma treatment method. The research in this paper is interesting and helpful to the sensors community and the presented results are good. The conclusion and references are also appropriate. However, the advantages compared with other publishing papers should be clarified further.

There are some comments as follows:

1, Some figures shown are unclear, such as Fig.2 and Fig.6. 

2, For table 2, please let the readers know how to measure the resistance.

3, In Fig.8, why do the samples have a difference, for example, samples 3 and 4.

Author Response

  1. (Fig. 2.) G stands for gauge length, W is the width of the narrow section, WO is the overall width, LO is the overall length, T is the thickness, D is the distance between grips, L is the length of the narrow section, R is the radius of fillet, and Wc is width at the center. The citation and ASTM standard are incorrectly marked in the previous submission and will be corrected. (Fig. 6.) 2D XRD was used to determine if AgCl crystals are present on the surface of magnesium-loaded PVA upon depositing the ink and before plasma treatment. To emulate the same conditions, we deposited a drop of silver nitrate ink onto the surface of the PVA until it turned cloudy, then removed the excess water and allowed the surface to dry. AgCl XRD peaks are known to be relatively sharp and distinct, but this approach failed to account for the difficulty of obtaining a clean XRD on a non-ideal substrate like the one we use to describe our system. Standard XRD procedure assumes a very flat and uniform sample, while the distribution of MgCl2 and presumably AgCl throughout the polymer is random. Therefore, the XRD also includes the PVA surface which is quite noisy due to the semicrystalline nature of PVA. Furthermore, it was not possible to subtract a background spectrum of PVA because this would mean comparing two different films. Because of these complications, we focused on the likelihood of each form of silver being present on the polymer surface based on possible reactions and the spectrum peaks. As shown in green, there are clear peaks in all of the areas that indicate AgCl, while other forms of silver have peaks that are not shown on the spectrum or are missing peaks shown on the spectrum. For quantification of the crystals formed, we hope to modify the XRD process or approach identification of the interface layer through alternative means in future work.
  2. Resistance was measured using a standard multimeter in which the probes are connected to a thin metal wire, which is then connected to the silver trace using a small amount of silver paste.
  3. The samples have slight differences in thickness dimension and structure, because the dogbones were cut from different areas of the same polymer sheet and solvent casting of PVA leads to slight variations in film thickness.

Reviewer 2 Report

In this manuscript, the authors report an inkjet printing method and a plasma treatment method using silver nitrate ink. The proposed method allows the fabrication of conductive silver electrodes without degrading the polymer substrate. It is also very impressive to achieve a sheet resistance of about 0.2 ohm/square under wet/dry and stretched/unstretched conditions.

The manuscript is well written in many parts, but it would be better if the following points were added.

Point 1: How reproducible is the resistivity of the fabricated silver electrode?

Point 2: I think the N number of the cycle test is small. Doesn't drift change occur in resistance value, etc.?

This manuscript is an important study on improving the quality of silver electrodes, and I believe that it is worthy of publication.

Author Response

  1. After ensuring that each sample was dried the same way before printing and plasma treated under the same conditions, we found that the sheet resistance of printed silver on two Mg-loaded PVA samples had negligible variation (0.2746 Ω/square vs. 0.2879 Ω/square), therefore the resistivity is also relatively constant. Because sheet resistance measurements are more time-intensive, we mostly report plain resistance values for comparison and found no significant variation in the printed lines of similar dimensions (~6-12 Ω) in at least 10 tested samples using a standard multimeter and thin metal wires/silver paste. These values align with our group's previously reported results on less reactive substrates such as PET, Kapton, printing paper, etc.
  2. Yes, we would like to increase the N cycle number for the wet strain measurements. Drift will most likely occur and where it occurs can provide valuable insight on how the silver chloride interface affects the device. However, it has been difficult to automate this process to extend to higher cycle numbers due to the difficulty of maintaining the same wet and dry states of the polymer while simultaneously measuring electrical resistance of a wet sample with reasonable precision. We hope to address this concern in future work by coupling a cyclic moisture apparatus with a cyclic strain device and making sure the strain/wetting does not interfere with electrical probe contact.